# Improving the Comprehension of Pathogenicity and Phylogeny in ‘*Candidatus* Phytoplasma meliae’ through Genome Characterization

**DOI:** 10.3390/microorganisms12010142

**Published:** 2024-01-11

**Authors:** Franco Daniel Fernández, Xiao-Hua Yan, Chih-Horng Kuo, Carmine Marcone, Luis Rogelio Conci

**Affiliations:** 1Instituto Nacional de Tecnología Agropecuaria (INTA), Centro de Investigaciones Agropecuarias (CIAP), Instituto de Patología Vegetal (IPAVE), Camino 60 cuadras km 5 ½ (X5020ICA), Córdoba X5020ICA, Argentina; 2Consejo Nacional de Investigaciones Científicas y Técnicas (CONICET), Unidad de Fitopatología y Modelización Agrícola (UFYMA), Camino 60 cuadras km 5 ½ (X5020ICA), Córdoba X5020ICA, Argentina; 3Institute of Plant and Microbial Biology, Academia Sinica, Taipei 115201, Taiwan; 4Department of Pharmacy, University of Salerno, 84084 Fisciano, Italy

**Keywords:** phytoplasma, genome, syntheny, effector protein, chinaberry, MPV, orthologs

## Abstract

‘*Candidatus* Phytoplasma meliae’ is a pathogen associated with chinaberry yellowing disease, which has become a major phytosanitary problem for chinaberry forestry production in Argentina. Despite its economic impact, no genome information of this phytoplasma has been published, which has hindered its characterization at the genomic level. In this study, we used a metagenomics approach to analyze the draft genome of the ‘*Ca.* P. meliae’ strain ChTYXIII. The draft assembly consisted of twenty-one contigs with a total length of 751.949 bp, and annotation revealed 669 CDSs, 34 tRNAs, and 1 set of rRNA operons. The metabolic pathways analysis showed that ChTYXIII contains the complete core genes for glycolysis and a functional Sec system for protein translocation. Our phylogenomic analysis based on 133 single-copy genes and genome-to-genome metrics supports the classification as unique ‘*Ca*. P. species’ within the MPV clade. We also identified 31 putative effectors, including a homolog to SAP11 and others that have only been described in this pathogen. Our ortholog analysis revealed 37 PMU core genes in the genome of ‘*Ca*. P. meliae’ ChTYXIII, leading to the identification of 2 intact PMUs. Our work provides important genomic information for ‘*Ca*. P. meliae’ and others phytoplasmas for the 16SrXIII (MPV) group.

## 1. Introduction

‘*Candidatus* Phytoplasma meliae’ (‘*Ca.* P. meliae’) has been associated with chinaberry (*Melia azedarach* L.) yellowing disease (ChTY) in Argentina [1] (subgroup 16SrXIII-G), Paraguay [2] (subgroup 16SrXIII-G), and Bolivia [3] (subgroup 16SrXIII-C). Affected chinaberry plants display characteristic symptoms such as reduced leaf size, yellowing, and witches’ broom. The mortality of the chinaberry, attributed to ‘*Ca*. P. meliae’ and the phytoplasma associated with chinaberry tree decline (ChTDIII, subgroup 16SrIII-B), poses a significant phytosanitary challenge, particularly in the northeast region of Argentina. This tree species is extensively cultivated in the area for furniture manufacturing [4]. The 16SrXIII group (Mexican periwinkle virescence; MPV) constitutes a monophyletic clade (Figure 1) within which twelve subgroups have been described so far (the last one being subgroup 16SrXIII-L, in plum trees) [5]. An interesting aspect of this group of phytoplasmas is their geographical distribution, which seems to be restricted to the American continent [1,6]. In addition, only a few host species have been associated with the MPV group, among them strawberries [7,8,9,10,11], potatoes [12] periwinkles [13], papayas [14], and broccoli [6], in addition to those already mentioned for ‘*Ca.* P. meliae’ above. Regarding vector insects, no species have thus far been described that could be involved in the dispersion of ‘*Ca*. P. meliae’ in Argentina or elsewhere in South America. However, the presence of a phytoplasma from the 16SrXIII group (a ‘*Ca*. P. hispanicum’-related strain) has been detected in the salivary glands of the sharpshooter *Homalodisca liturata* Ball [15]. Nevertheless, studies confirming the transmission capability of these phytoplasmas have not been conducted. This context underscores the need to further advance our understanding of the biology of this specific clade of phytoplasmas.

Thanks to the reduction in the cost of sequencing over the last decade, phytoplasma sequencing projects have notably increased. These bacteria have a unique biology, and genomic knowledge has allowed us to understand aspects of pathogenicity and evolution that have never been described [16,17,18,19]. Currently, no genomic data have been generated for ‘*Ca.* P. meliae’, which makes it difficult to study the mechanisms associated with their pathogenicity, evolution, or dispersion (mediated by insects). In this study, we report a draft genome of the ‘*Ca.* P. meliae’ strain ChTYXIII-Mo (subgroup 16SrXIII-G), obtained from infected chinaberries in Argentina. The goal of this study was to provide basic genomic information in order to understand fundamental aspects of the evolution and pathogenicity of this phytoplasma and others related to the MPV clade.

## 2. Materials and Methods

### 2.1. Plant Samples

The ‘*Ca.* P. meliae’ strain ChTYXIII-Mo (KU850940) [1] was maintained under controlled conditions in greenhouses and propagated in chinaberry tree plantlets through grafting. Total genomic DNA was extracted from infected midribs using the DNeasy Plant Mini Kit (Qiagen, Germany) following the manufacturer’s instructions. Quality and quantity controls were assessed using electrophoresis in 1% agarose gels and spectrophotometry (Nanodrop-1000).

### 2.2. Library Construction and Sequencing

Total DNA was used to construct paired-end libraries (150 bp) according to the TruSeq™ DNA Nano protocol and sequenced on the Illumina Novaseq platform (Macrogen, Republic of Korea). The quality of the raw reads was assessed using FastQC (https://github.com/s-andrews/FastQC) and then trimmed using Trim Galore (https://github.com/FelixKrueger/TrimGalore) with default settings.

### 2.3. Assembly and Annotation

A metagenomic approach was implemented for assembly based on previous pipelines, with some modifications [20]. Trimmed reads were assembled using Unicycler (bridging mode: normal) with Spades correction and Pilon polish [21]. Assembled contigs belonging to phytoplasmas were identified using BLASTx (E = 1 × 10^−20^, word size = 11) against a local database constructed using phytoplasma genome sequences available from NCBI (txid33926). Trimmed reads were mapped to phytoplasma-assigned contigs using Bowtie2 v2.3.4.3 (default parameters) [22], and an iterative process was used until the assembly was completed. The completeness of the draft assembly was estimated using BUSCO [23] and CheckM [24]. The final draft genome was annotated using the NCBI Prokaryotic Genome Annotation Pipeline [25]. The KAAS–KEGG Automatic Annotation Server (https://www.genome.jp/kegg/kaas/) was used for the functional characterization of protein-coding regions and the reconstruction of metabolic pathways.

### 2.4. Identification of Putative Effector Proteins

Putative effector proteins were identified using previously established pipelines [20,26]. Signal peptides were predicted using the Signal IP v5.0 server (https://services.healthtech.dtu.dk/service.php?SignalP-5.0), and the criteria defined in [20] were applied to define positive candidates. Proteins that passed this filter (signal peptide+) were analyzed using the TMHMM-2.0 server (https://services.healthtech.dtu.dk/service.php?TMHMM-2.0), and those without any transmembrane helices domains after the signal peptides were selected were considered as putative secreted proteins (PSPs). The PSPs were then analyzed using the Conserved Domains Database search tool (www.ncbi.nlm.nih.gov/Structure/cdd/wrpsb.cgi, E-value = 0.01), and nuclear signal prediction was carried out using NLStradamus (http://www.moseslab.csb.utoronto.ca/NLStradamus/). The subcellular localization was determined using the LOCALIZER (http://localizer.csiro.au/) and ApoplastP (http://apoplastp.csiro.au/) servers. The final set of proteins [signal peptide (+); transmembrane domains outside SP (−)] were subjected to reciprocal BLASTp searches (E-value ≤ 1 × 10^−5^) against aster yellows witches’ broom (AYWB) phytoplasma proteins (taxid: 322098) to identify homologous SAPs [27]. 

### 2.5. PMU Core Gene and Intact PMU Identification

The PMU gene information was based on the homologs of eight core genes (*tra5*, *dnaB*, *dnaG*, *tmk*, *hflB*, *himA*, *ssb*, and *rpoD*) as previously defined in [26]. To conduct a comparative analysis with the other phytoplasma genomes employed in this study, homologous PMU core gene clusters were identified using OrthoMCL v1.3. The prerequisites for an intact PMU are as follows: (1) An intact PMU should encompass a size of approximately 10 kb. (2) The intact PMU must incorporate no less than four distinct PMU core genes in the same orientation, excluding the *tra5* gene. A graphical representation of PMU comparisons among phytoplasma genomes was generated using Clinker v0.0.27 [28].

### 2.6. Ortholog Clustering and Phylogenetic Analyses

Orthologous protein clusters were identified using OrthoFinder v2.5.2 (https://github.com/davidemms/OrthoFinder). Genome sequences of representative ‘*Candidatus* Phytoplasma species’ (‘*Ca.* P. species’) were obtained from GenBank (Appendix A). For phylogenetic analyses, concatenated nucleotide sequences of single-copy core genes or single genes were aligned using MAFFT v7.450 in Geneious R.10 software (Biomatters Ltd., Auckland, New Zealand). Phylogenetic trees were constructed using IQ-TREE (http://www.iqtree.org/) with an automatic substitution model and by conducting an ultrafast bootstrap analysis with 1000 replicates. Whole-genome comparisons were performed using fastANI v1.1 [29]. Additionally, we utilized the genome-to-genome distance calculator online tool (https://ggdc.dsmz.de/home.php) to calculate dDDH values.

## 3. Results and Discussion

### 3.1. Assembly and Key Features of the Draft Genome of ‘Ca. Phytoplasma meliae’

The genome sequencing of the ‘*Ca*. P. meliae’ strain ChTYXIII generated approximately 3.5 Gbp of raw reads (NCBI accession: PRJNA638346), providing ~97-fold coverage of the draft genome. The assembly resulted in 21 contigs totaling 751,949 bp with a G + C content of 27.31% (Table 1) (Figure 2). Recently, the draft genome of Phyllody Phytoplasma StrPh-CL (subgroup 16SrXIII-F) was reported, which has been implicated in causing strawberry phyllody in Chile [30]. The authors of this study assigned a genome completeness value of approximately 90% to StrPh-CL based on PFGE experiments. Therefore, we can assume that the genome completeness values of ‘*Ca*. P. meliae’ are likely to be similar, since both phytoplasmas are phylogenetically closely related (belonging to group 16SrXIII) (Table 1). In silico completeness metrics showed values of 97.34%, along with a possible contamination value of 3.29% (CheckM), and 94.70% with 413 complete and single-copy BUSCO groups out of 151 total BUSCO groups from the mollicutes_odb10 lineage. However, as discussed in previous works [20], these estimates are based on a limited number of marker genes, and caution should be exercised when interpreting them due to the limited representation of phytoplasma genomes in current databases.

The 21 contigs composing the draft assembly ranged from 1832 bp to 137,693 bp (N50 = 53,850). In the annotation process, 669 CDSs (full-length coding sequences) were identified, with 472 annotated as proteins with assigned functions and 197 annotated as hypothetical proteins. One operon for rRNA genes and 34 tRNAs were also identified (Table 1). The functional annotation of CDSs using BlastKOALA (https://www.kegg.jp/blastkoala/) assigned 387 of 669 CDSs (~58%) to orthologs in the KEEG database. Of the 299 KO categories, 240 were described with only 1 gene, while the remaining 59 presented more than 1 copy, representing 147 genes (~21% of total CDSs). The proportion of multicopy genes in the ‘*Ca*. P. meliae’ ChTYXIII genome (~21%) was higher than that observed in other related phytoplasmas such as ‘*Ca*. P. hispanicum’ StrPh-CL (11%), ‘*Ca*. P. solani’ SA-1 (18.5%) [20], ‘*Ca*. P. asteris’ AY-WB (10.2%), [26], ‘*Ca*. P. asteris’ OY-M (14.1%) [16], or ‘*Ca*. P. australiense’ PAa (12.1%) [31]. Multicopy genes in PMU-like regions and genome size appear to be positively correlated with a broad host range in phytoplasmas [20]. However, despite being associated with only two hosts, chinaberry [1,3] and plum [5], the total number of hosts for ‘*Ca*. P. meliae’ may have been underestimated since little information is available regarding its presence in native species such as weeds and its vector insects remain unknown. A more comprehensive understanding of the host range of this phytoplasma is necessary. 

### 3.2. Metabolic Pathways

The ‘*Ca.* P. meliae’ draft genome contains 184 CDSs assigned to Metabolism, 236 CDSs assigned to Genetic Information Processing, and 54 CDSs assigned to Signaling and Cellular Processes (Appendix A). Transporter membrane proteins play fundamental roles in phytoplasma metabolism, allowing for the incorporation of metabolites and contributing to protein secretion in the host cell’s cytoplasm. Among these, the ATP-binding cassette (ABC) transporters form one of the largest known protein families and are widespread in bacteria, archaea, and eukaryotes. These proteins are best known for their role in the importation of essential nutrients and the exportation of toxic molecules, but they can also mediate the transport of many other physiological substrates [32]. We identified 28 ABC transporter genes in the ‘*Ca.* P. meliae’ genome, including the complete pathway for spermidine/putrescine transport (*potA*, *potB*, *potC*, and *potD*), lysine transport (*lysX1*, *lysX2*, and *lysXY*), and Zinc/Manganese/Iron (II) transport (*troA*, *troC*, *troD*, and *troB*) (Appendix A). We also identified genes for the protein translocation system (Sec system), including *secA*, *secE*, *secY*, *ffh*, *ftsY*, *yidC*, *dnaJ*, *dnaK*, *grpE*, and *groEL*, which suggest a functional Sec system in ‘*Ca.* P. meliae’. The deduced amino acid sequence for the antigenic membrane protein (AMP) homolog (ChTY_001350) was 158 aa, with an estimated molecular weight of 17.07 kDa also being found. A Phyto-Amp conserved domain (Cdd: pfam15438) was also recorded in the interval 1–103 (E = 0.02), supporting the notion that this protein is an ortholog of a previously described antigenic membrane protein (Amp) [33]. 

DNA replication, reparation, and duplication gene sets were identified in the draft genome, as described in the genomes of related species such as ‘*Ca*. P. solani’ strains SA-1 and 284/9, the ‘*Ca*. P. australiense’ strain PAa, and the ‘*Ca*. P. hispanicum’ strain StrPh-CL. The annotation process allows us to identify excision repair complex uvrABC, DNA polymerase III (subunits alpha, beta, delta, delta’, gamma/tau), ruvX (Holliday junction resolvase-like protein), and the recA gene. These findings shed light on the molecular mechanisms involved in DNA maintenance and repair, providing valuable insights into the genomic adaptations and survival strategies of ‘*Ca.* P. meliae’.

Within the carbohydrate metabolism, we found the core module for glycolysis (genes *gapA*, *pyk*, *pgk*, *eno*, *tpiA*, and *gpmI*) and pyruvate oxidation (genes *pdhA*, *pdhB*, *pdhD*, and *aceF*), which supports the idea that ‘*Ca.* P. meliae’ could depend on glycolysis for energy generation. This pathway has been described as the major energy-yielding pathway for phytoplasmas [34]. Finally, we identified the complete Open Reading Frame (ORF) for the gene sucrose phosphorylase (*gtfA*, CHTY_000510), which could provide an adaptation to plant environments by cleaving imported sugars [31]. 

### 3.3. Secreted Protein Identification Revealed a Unique Effectors Repertoire

In the draft genome of ‘*Ca.* P. meliae’, we identified 31 putative secreted proteins (Appendix A). A BLASTp search against the SAP protein repertoire of the AYWB phytoplasma (accession CP000061) showed the presence of putative orthologs for SAP72 (53.31%), SAP41 (52.29%), SAP68 (76.00%), SAP67 (59.67%), SAP21 (38.10%), and SAP11 (50.00%), but not for SAP54, SAP05, and the TENGU factors. The SAP11 homolog consisted of 116 aa (CHTY_003225) with a predicted signal peptide domain (score = 0.98, position 1–32 aa), a characteristic SMV signal (pfam12113, position 1-33 aa, E-value = 5.71 × 10^−8^), a nuclear signal localization (position 35–49 aa), and coiled-coil domains (position 84–106aa), features compatible with those described for the SAP11 homolog in ‘*Ca*. P. asteris’ AYWB (Figure 3A) and in other diverse phytoplasma taxons.

Additionally, we found a conserved synteny in the genomic context of SAP11 with homologous regions in the chromosome of ‘*Ca.* P. asteris’ AYWB and *‘Ca.* P. ziziphi’ Jwb-nky (Figure 3B), suggesting a possible horizontal transfer. *A. thaliana* transgenic lines expressing SAP11 displayed curly leaves and an increased number of axillary stems that resemble the witches’ broom symptoms exhibited by the AY-WB phytoplasma [35,36]. Greenhouse chinaberry plantlets infected by ‘*Ca.* P. meliae’ exhibit typical witches’ broom symptoms, while those infected with ‘*Ca.* P. pruni’ strain ChTDIII (16SrIII-B) show symptoms of yellowing and the shortening of internodes but not witches’ broom (Appendix A). This could suggest the presence of an SAP11 homolog-associated mechanism in the generation of witches’ broom symptoms, since no SAP11 homologs were described for the ChTDIII phytoplasma.

Furthermore, a BLASTp search revealed that 10 out of the 31 putative secreted proteins appear to be exclusive to ‘*Ca.* P. meliae’, as no homologous sequences were identified in the gene database. Among these proteins are CHTY_001115 (115aa), which exhibits a domain associated with a multidrug resistance efflux pump (cl34307), and CHTY_002595 (227aa), which possesses a peptidase-related domain. Furthermore, CHTY_002535 (391aa) was predicted to localize in the chloroplast (score = 0.991, position 105–125) and features a substrate-binding domain similar to that of an ABC-type nickel/oligopeptide-like import system (cl01709). These proteins serve as compelling targets for investigation, as they hold the potential to provide valuable insights into the distinctive pathogenicity mechanisms employed by this pathogen.

### 3.4. PMU Identification

Ortholog analysis revealed that the genome of ‘*Ca*. P. meliae’ ChTYXIII contains 37 PMU core genes (Figure 4). Using the information on PMU core genes, two intact PMUs were identified within the ‘*Ca*. P. meliae’ ChTYXIII genome: ChTYXIII_PMU01 (JACAOD020000005: 60,425–68,641 bp) (Figure 5A) and ChTYXIII_PMU02 (JACAOD020000003: 1–6117 bp) (Figure 5B). Notably, although ChTYXIII_PMU02 comprises only three PMU core genes, it exhibits similarities to the PMUs found in other phytoplasma genomes.

Previous studies have categorized intact PMUs into three primary types (A, B, and C) based on the presence/absence and organization of PMU core genes [37]. Among phytoplasmas, type A PMUs, characterized by the presence of *tmk* upstream of *dnaB*, have the broadest distribution. Conversely, type B PMUs, where *tmk* is positioned downstream from *dnaB*, are predominantly confined to ‘*Ca*. P. australiense’. Following this classification, ChTYXIII_PMU01 is categorized as a Type B PMU, while ChTYXIII_PMU02 falls under the category of a Type A PMU.

Our synteny analysis indicated that ChTYXIII_PMU01 exhibited conservation with PMUs from ‘*Ca*. P. australiense’ NZSb11 and PAa. Interestingly, ChTYXIII_PMU01 harbored a copy of SAP36 (locus tag: CHTY_001500) (Figure 5A), a feature not found in the other PMUs. The entire sequence of contig JACAOD020000003.1 was classified as a PMU (ChTYXIII_PMU02). The PMU core genes of ChTYXIII_PMU02 demonstrated synteny conservation with the PMU of the closely related phytoplasma strain ‘*Ca*. P. hispanicum’ StrPh-Cl, as well as a more distantly related strain, ‘*Ca*. P. solani’ SA-1. No SAPs or putative secreted proteins were identified within the PMU region (Figure 5B).

As previously described, the majority of sequenced phytoplasmas typically have only one type of PMU [37]. In the case of ‘*Ca*. P. meliae’ ChTYXIII, two types (A and B) have been described, such as phytoplasmas like ‘*Ca*. P. australiense’ PAa and ‘*Ca*. P. solani’ SA-1, which are more closely related phylogenetically.

### 3.5. Phylogenomic Analysis and Taxonomic Delineation

Comparative genomics based on ortholog identification between draft genomes of ‘*Ca.* P. meliae’ ChTYXIII and the representative genome sequence of 11 ‘*Ca*. P. species’ (Appendix A) and *A. palmae* (accession FO681347) reveals the presence of 132 single-copy genes common to all species (Appendix A). Phylogenetic reconstruction based on our analysis of 132 single-copy genes demonstrated that ‘*Ca.* P. meliae’ exhibits a close relationship with StrPh-Cl, as both have been grouped in the same cluster (bootstrap = 100) (Figure 6A,B). The same clustering patterns could be reconstructed when analyzing the phylogeny of the individual single-copy genes *secA*, *secY*, and *tuf* (Appendix A). Phytoplasmas belong to group 16SrXIII, which constitutes a well-supported branch that represents a distinct lineage compared to representative members of previously described ‘*Ca*. P.’ species, suggesting that the strains within group 16SrXIII share a common ancestor [38] with a restricted distribution to the Americas. So far, eleven subgroups have been described (16SrXIII-A, B, C, D, E, F, G, H, J, K, and L) [7], and two ‘*Ca.* P.’ species have been proposed: ‘*Ca.* P. hispanicum’ [38] and ‘*Ca.* P. meliae’ [1]. 

The description of *‘Ca.* P. meliae’ was conducted by analyzing the 16S rRNA gene, *secA*, and ribosomal protein genes. On the other hand, ‘*Ca.* P. hispanicum’ was primarily based on the 16S rRNA gene. However, the biological characteristics of both phytoplasmas, such as symptomatology in vinca, host range, and geographic distribution, were also taken into consideration. The representative sequences of the 16S rRNA gene within the 16SrXIII subgroups exhibit similarity values exceeding 98.65%. This suggests the need to expand the characterization of other genes, genome-scale analyses, and additional biological properties to achieve a species-level description.

In 2022, revisions were made to the general guidelines for naming new ‘*Ca.* P.’ species [39]. These revisions recommended the use of the whole-genome average nucleotide identity (ANI) metric as additional support for the traditional classification based on the 16S rRNA gene. ANI calculates the average nucleotide identity of all regions shared between two genomes, providing a robust resolution for distinguishing strains within the same or between closely related species [29]. To evaluate the taxonomic classification of ‘*Ca.* P. meliae’, ANI values were calculated against available phytoplasma genomic sequences in the NCBI database (https://www.ncbi.nlm.nih.gov/datasets/genome/?taxon=33926). None of the evaluated sequences showed ANI values higher than 95%. The phytoplasma strain StrPh-Cl (16SrXIIII-F) [30] exhibited the highest ANI value of 92.76% in the comparisons (Appendix A). These findings, combined with the species’ restricted geographical distribution and specific biological properties, provide further support for the taxonomic classification of ‘*Ca.* P. meliae’.

As the repertoire of available phytoplasma genomes continues to expand, comparative studies that more accurately represent reality can be conducted. In the case of the 16SrXIII group, there is currently genomic information for only 2 out of the 11 described subgroups (16SrXIII-G and 16SrXIII-F), limiting diversity studies to the individual sequences of a few genes. This study describes the partial genome of ‘*Ca.* P. meliae’, which represents a valuable genomic resource for analyzing diversity, evolution, and pathogenicity mechanisms within the 16SrXIII group.

## 4. Conclusions

This study presents the draft genome of the ‘*Ca*. P. meliae’ strain ChTYXIII. Through functional analyses, we have unveiled highly conserved genes involved in processing and metabolism, a pattern commonly observed among phytoplasmas. Notably, despite ‘*Ca*. P. meliae’ being associated with only a few known host species, we have identified numerous multicopy genes. Additionally, our investigation has led to the discovery of several putative effector proteins with intriguing characteristics, alongside the well-known SAP11 homolog. These findings suggest that these effector proteins may play pivotal roles in regulating pathogenicity mechanisms.

The genomic-based phylogenetic analysis not only supports the taxonomic classification of ‘*Ca.* P. meliae’ but also emphasizes the importance of expanding the genomic repertoire within this group of pathogens. Specifically, our ortholog analysis revealed 37 PMU core genes in the genome of ‘*Ca*. P. meliae’ ChTYXIII, leading to the identification of 2 intact PMUs. Although ChTYXIII_PMU02 comprises only three PMU core genes, its similarities to PMUs found in other phytoplasma genomes raise intriguing questions about the functional diversity of these genetic elements.

This comprehensive analysis contributes not only to our understanding of ‘*Ca*. P. meliae’ but also enhances our broader knowledge of phytoplasma genomics. It underscores the need for the continued exploration and expansion of genomic studies within this diverse group of plant pathogens.

## Figures and Tables

**Figure 1 microorganisms-12-00142-f001:**
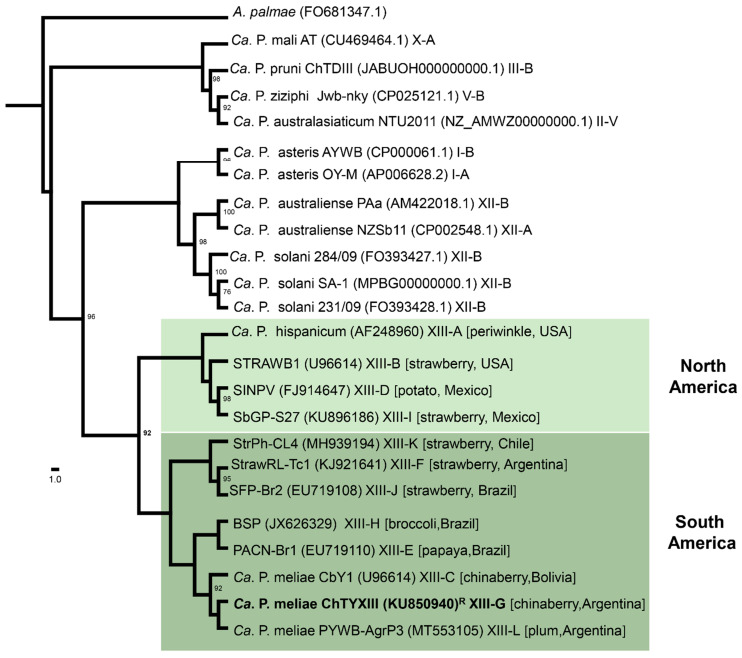
Maximum-likelihood phylogeny of representative phytoplasmas based on 16Sr RNA sequences using IQ-TREE (http://www.iqtree.org/, accessed on 1 October 2023). The sampling emphasizes the 16SrXIII group (Mexican periwinkle virescence; MPV); *Acholeplasma palmae* was used as an outgroup. The numbers on the branches are bootstrap values (expressed as percentages of 1000 replicates); only values > 70% are shown. The GenBank accession number for each taxon is in parentheses, and the 16Sr group/subgroup classification is also provided. The MPV clade is highlighted with green boxes according to geographic distribution, and the ‘*Ca.* P. meliae’ strain sequenced in this study is in bold. R: reference strain.

**Figure 2 microorganisms-12-00142-f002:**
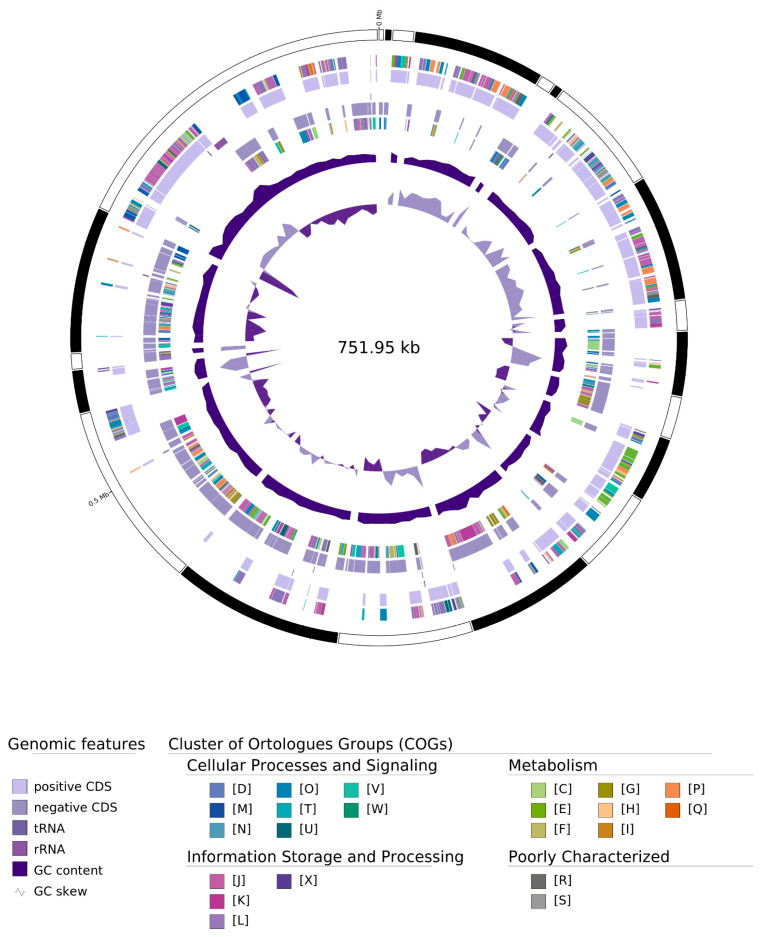
Circular map representation of the ‘*Ca.* P. meliae’ draft genome. Labeling from outside to inside: Contigs, COGs on the forward strand, CDS, tRNAs, RNAs on the forward strand, CDS, tRNAs, RNAs on the reverse strand, COGs on the reverse strand, GC content, and GC skew.

**Figure 3 microorganisms-12-00142-f003:**
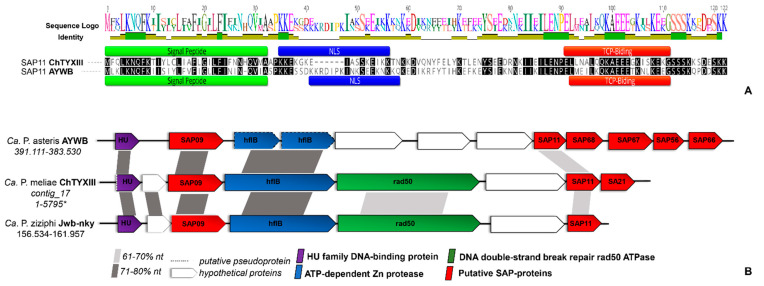
Analysis of the ‘*Ca.* P. meliae’ SAP11 homolog. (**A**) Alignment of the amino acid sequence of the SAP11 effector in ‘*Ca.* P.asteris ’(AYWB) and the homolog identified in the genome of ‘*Ca*. P. meliae’ (ChTYXIII) using MAFFT v7.450. The signal peptide motifs (green), nuclear localization signal (blue), and TCP-binding domain (red) are shown. (**B**) The syntenic organization of contig ChTYXIII_17, which contains the SAP11 homolog. The genomic localization (start–end) is given below the ‘*Ca*. P. specie’ identification. SAP homologs are shown in red. Nucleotide sequence similarities between conserved regions are illustrated by different shades of gray. * complete contig.

**Figure 4 microorganisms-12-00142-f004:**
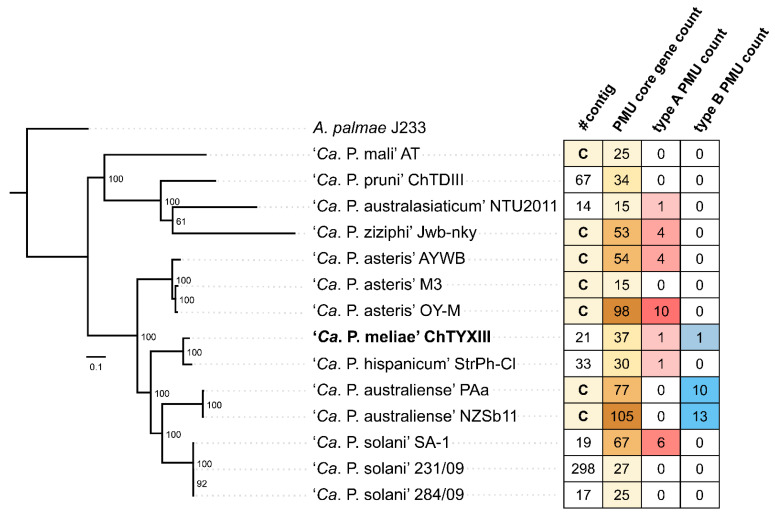
The relationships between the PMUs of the phytoplasma species used in this study. The information to the right shows the number of contigs from the gb file (C indicates complete genome), the number of copies of the PMU core gene, the number of intact type A PMUs, and the number of intact type B PMUs.

**Figure 5 microorganisms-12-00142-f005:**
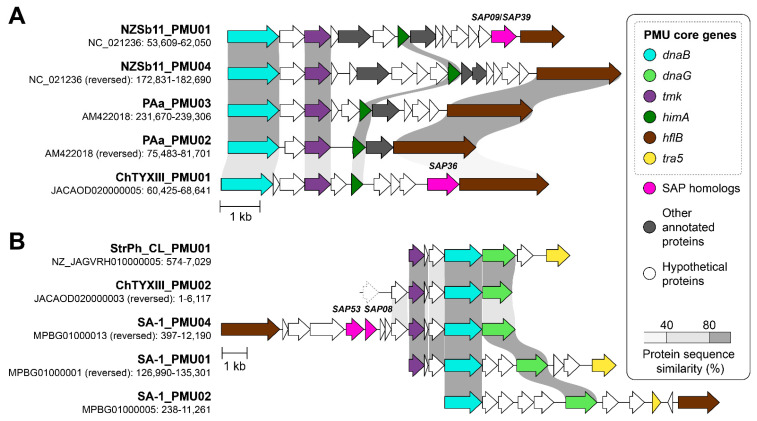
Comparison of PMUs. The genomic location (contig: start–end) of each PMU is labeled. Genes that are commonly associated with PMUs are highlighted in color; SAPs are colored in pink. Protein sequence similarities between conserved regions of PMU core genes are illustrated by different shades of gray. Partial genes are drawn with dotted borders. (**A**) ChTYXIII_PMU01 with other type B PMUs. (**B**) ChTYXIII_PMU02 with other type A PMUs.

**Figure 6 microorganisms-12-00142-f006:**
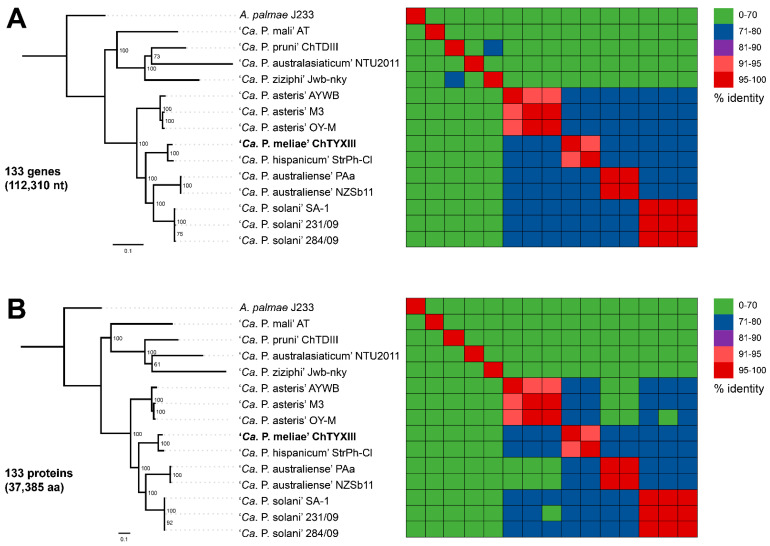
Molecular phylogeny based on the nucleotide (model GTR + F + I + G4) (**A**) and amino acid (LG + F + I + G4) (**B**) sequences of 133 core genes. *Acholeplasma palmae* was included as an outgroup to root the tree. The numbers on branches indicate the level of bootstrap support (1000 replicates). The scale bar represents the number of substitutions per site. The heatmaps on the right-hand side are colored based on sequence identity. In bold the draft genome obtained in this work.

**Table 1 microorganisms-12-00142-t001:** Genome statistics of ‘*Ca*. Phytoplasma meliae’ and related ‘*Ca*. Phytoplasma species’.

	*Ca*. meliae	*Ca.* hispanicum	*Ca*. Solani	*Ca*. australiense	*Ca*. asteris
Features/Strain	ChTYXIII	StrPh-CL	SA-1	PAa	AYWB
# Contigs	21	33	19	1	1
Total Length (bp)	751.949	627.584	821.322	879.324	706.569
G + C content (%)	27.31	25.40	28.30	27.00	27.00
N(50)	53.850	82.058	76.256	-	-
Assembly status	draft	draft	draft	complete	complete
Total CDSs	669	559	709	839	671
CDSs w-function	472	440	452	502	450
CDSs h-protein	197	119	257	337	221
rRNA-operons	1	2	2	2	2
tRNA	34	32	32	35	32
BUSCO	94.70%	93.40%	95.40%	94.70%	93.40%
Genbank #	JACAOD000000000.2	JAGVRH000000000.1	MPBG01000000.1	AM422018.1	CP000061.1

## Data Availability

RAW reads were deposited in NCBI Sequence Read Archive (SRA) under the accession SRX10975989. The de novo *genome* draft assembly of ChTYXIII was deposited in GenBank under the accession JACAOD000000000.2 (BioProject: PRJNA638346, BioSample: SAMN15186628). The aligned sequence datasets used for phylogenomic analysis were deposited in Zenodo repository https://zenodo.org/doi/10.5281/zenodo.10459693.

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
