# Peer review of "Improving the Comprehension of Pathogenicity and Phylogeny in ‘Candidatus Phytoplasma meliae’ through Genome Characterization"

_microorganisms, 2024, doi:10.3390/microorganisms12010142_

Round 1
Reviewer 1 Report
Comments and Suggestions for Authors
The paper from Fernandez et al. deals with the very interesting and relevant topic of unknown phytoplasma genome characteristics. Research studies like this provide very useful knowledge that can be used to gain additional knowledge on the subject of phytoplasma epidemiology, phytoplasma effector proteins, and overall pathogenicity factors. The paper is clear and presented in a well-structured manner. The manuscript is scientifically sound, and the experimental design is appropriately chosen. The introduction is clear and contains all the necessary information. Figures and tables are appropriate, and the data is scientifically sound. The results are interpreted appropriately and consistently throughout the manuscript. Conclusions are consistent and in agreement with the evidence and arguments presented.
Overall, it is a well-executed research flow that provides sufficient novelty. It is a step in the right direction, but only if it is followed by research on insect vectors and natural plant reservoirs. Only then will the epidemiology of any given phytoplasma be fully understood.
Few additional comments:
- It would be useful to include a brief overview of insect vectors and natural plant reservoirs of this phytoplasma (if any are known), perhaps accompanied with knowledge of the phylogenetically closest phytoplasma, e.g., Ca. P. hispanicum, in the introduction section. Because both phytoplasmas have a limited geographical distribution, any local information would be useful to scientists from other continents, especially since MPV-clade phytoplasmas are phylogenetically related to those from distant geographical regions.
- Please provide information on the software used for the maximum-likelihood phylogeny in Figure 1. Since this figure is in the introduction part, it would be useful to add this information to the figure caption.
- Line 23 in Abstract - Please change “Ca. P.' specie“ to “Ca. P.' species” - species refers to both singular and plural.
- Line 229 in caption for Figure 3 – same comment as above.
Author Response
Response to reviewer #1
Dear Revisor, Thank you very much for taking the time to review this manuscript. Please find the detailed responses below and the corresponding revisions/corrections:
Coment_1: It would be useful to include a brief overview of insect vectors and natural plant reservoirs of this phytoplasma (if any are known), perhaps accompanied with knowledge of the phylogenetically closest phytoplasma, e.g., Ca. P. hispanicum, in the introduction section. Because both phytoplasmas have a limited geographical distribution, any local information would be useful to scientists from other continents, especially since MPV-clade phytoplasmas are phylogenetically related to those from distant geographical regions.
Response_1: Thank you for the suggestion provided. Unfortunately, we have not initiated nor do we have data on natural hosts or insect vectors for ‘Ca. P. meliae’. Regarding the information available on ‘Ca. P. hispanicum’, we only cite the work by Servín-Villegas et al., Int J Syst Evol Microbiol 2018;68:2093–2101 DOI 10.1099/ijsem.0.002745, in which the presence of phytoplasmas from the 16SrXIII group in insects of the species Homalodisca liturata is described. Although this work does not provide evidence of the transmission capacity of said phytoplasma, we will consider including it in the text while acknowledging these limitations.
Coment_2: Please provide information on the software used for the maximum-likelihood phylogeny in Figure 1. Since this figure is in the introduction part, it would be useful to add this information to the figure caption.
Response_2: Thank you for pointing this out. We agree with this comment. Information regarding software used iin phylogenetic reconstruction was added in the Figure 1 Caption
Coment_3: Line 23 in Abstract - Please change “Ca. P.' specie“ to “Ca. P.' species” - species refers to both singular and plural.
Response_3: Thank you for pointing this out. We agree with this comment. We change “Ca. P.' specie“ to ‘Ca. P. species’ in the main text
Coment_4: Line 229 in caption for Figure 3 – same comment as above.
Response_4: Thank you for pointing this out. We agree with this comment. Information regarding software used iin phylogenetic reconstruction was added in the Figure 3 Caption
Reviewer 2 Report
Comments and Suggestions for Authors
This paper reports the first full genome assembly for Ca. Phytoplasma meliae and provides comparative analysis of genome content, providing important information that can be used for further study of mechanisms for host interactions and pathogenicity in this economically important group of bacterial parasites. Overall the paper is well done, with excellent illustrations. My only suggestions for improvement are as follows:
1. Please provide a more detailed description of the phylogenetic analysis. Were different substitution models used for different data partitions (e.g., individual genes) in the concatenated analysis? Did this analysis include 16S in addition to protein-coding genes? A list of genes included should be provided as a supplemental document.
2. Please deposit the aligned sequence dataset used for phylogenetic analysis in a public archive where it is available to other researchers and provide a link in the paper. This is important for repeatability.
Author Response
Response to reviewer #2
Dear Revisor, Thank you very much for taking the time to review this manuscript. Please find the detailed responses below and the corresponding revisions/corrections:
Coment_1. Please provide a more detailed description of the phylogenetic analysis. Were different substitution models used for different data partitions (e.g., individual genes) in the concatenated analysis? Did this analysis include 16S in addition to protein-coding genes? A list of genes included should be provided as a supplemental document.
Response_1: Thank you for your valuable feedback. In our study, the substitution models for both amino acids (aa) and nucleotides (nt) were chosen based on the Bayesian Information Criterion (BIC). The specific models selected for the phylogenetic trees of concatenated genes are detailed in the figure caption. It's important to note that in our analysis, we exclusively utilized protein-coding genes. Additionally, we have included a supplemental document, Table Supplement 4, which provides a list of the genes used for constructing the phylogenetic tree. We hope this additional information addresses your inquiry regarding the phylogenetic analysis.
Coment_2: Please deposit the aligned sequence dataset used for phylogenetic analysis in a public archive where it is available to other researchers and provide a link in the paper. This is important for repeatability.
Response_2: Thank you for bringing this to our attention. We appreciate the importance of data accessibility. Aligned sequence dataset used for phylogenomic analysis were deposited in Zenodo repository https://zenodo.org/doi/10.5281/zenodo.10459693. This information was added in the Data availability section.
Reviewer 3 Report
Comments and Suggestions for Authors
The manuscript with the title “Improving the comprehension of pathogenicity and phylogeny in 'Candidatus Phytoplasma meliae' through genome characterization” provides first genome information on the causal agent of phytoplasma in chinaberry in Argentina. Using meta-genomics approach the authors analyzed the draft genome of 'Ca. P. meliae' strain ChTYXIII, providing the first description.
The abstract is a providing a good summary of the manuscript.
Introduction
In the first paragraph, chinaberry tree shall be identified by the scientific name once (to ensure international audience precisely understood which plants are affected), afterwards using the common name is fine.
Material and method
Provides sufficient details and has the appropriate subheadings.
Results
The subheadings are consistent with the findings from the paragraphs and are in line with the objectives of the study.
Conclusions
The authors reached their objectives in their study and managed to draft the genome of Phytoplasma infecting chinaberry.
References
Out of 47 sources, a few are from recent years (latest are 3 papers from 2022). But overall all the reference list is citing relevant works for the study their conducted.
Best regards.
Comments on the Quality of English Languagesome grammar and syntax improvement could increase readability.
Author Response
Dear Revisor, Thank you very much for taking the time to review this manuscript. Please find the detailed responses below and the corresponding revisions/corrections:
Coment_1: In the first paragraph, chinaberry tree shall be identified by the scientific name once (to ensure international audience precisely understood which plants are affected), afterwards using the common name is fine.
Response_1: Thank you for pointing this out. We agree with this comment. We use the scientific name in the Introduction. The common name Chinaberry was used in the rest of the manuscript.
Reviewer 4 Report
Comments and Suggestions for Authors
In the current manuscript, Fernández and co-authors used a meta- 17 genomics approach to analyze the draft genome of 'Ca. P. meliae' strain ChTYXIII. Overall work presented in the manuscript is interesting. Experiments are planned and executed well. Results are also well discussed. However, there are some grammatical and typographical mistakes. I will recommend manuscript be considered for publication after minor revision.
Authors can consider the following comments while improving the present draft of a manuscript,
1. Line-31: Please provide the full name of the pathogen at the beginning.
2. Line-34-37: “China tree” is this right? Please rewrite the whole sentence, as it is too big and not clear.
Comments on the Quality of English LanguageEnglish is acceptible.
Author Response
Response to reviewer #4
Dear Revisor, Thank you very much for taking the time to review this manuscript. Please find the detailed responses below and the corresponding revisions/corrections:
Coment_1: Line-31: Please provide the full name of the pathogen at the beginning.
Response_1: Thank you for pointing this out. We agree with this comment. We provide the full name of the pathogen
Coment_2: Line-34-37: “China tree” is this right? Please rewrite the whole sentence, as it is too big and not clear.
Response_2: Thank you for the comment. We agree with this suggestion. We corrected China tree by Chinaberry and and have reformulated the entire sentence to enhance clarity.